Effects of artificially-simulated acidification on potential soil nitrification activity and ammonia oxidizing microbial communities in greenhouse conditions

Zhang Xiaolan
Shan Xuan
Fu Hongdan fuhongdan@syau.edu.cn
Sun Zhouping sunzp@syau.edu.cn
Shenyang Agricultural University , Shenyang , China
Khan Amanullah
Electronic publication date: 2022 Oct 3
Publication date: 2022
Volume: 10
Electronic Location ID: e14088
Received 2022 May 20; Accepted 2022 Aug 29
Copyright: © 2022 Zhang et al.
Copyright year: 2022
Copyright holder: Zhang et al.
License: This is an open access article distributed under the terms of the Creative Commons Attribution License, which permits unrestricted use, distribution, reproduction and adaptation in any medium and for any purpose provided that it is properly attributed. For attribution, the original author(s), title, publication source (PeerJ) and either DOI or URL of the article must be cited.
License URL: https://creativecommons.org/licenses/by/4.0/

Keywords: Simulated acidification, Nitrification, Ammonia oxidizers, Abundance, Community structure, Tomato

Funding: National Natural Science Foundation of China 31902093 China Agriculture Research System (CARS-23) This work was supported by the National Natural Science Foundation of China (Grant No. 31902093) and the China Agriculture Research System (CARS-23). The funders had no role in study design, data collection and analysis, decision to publish, or preparation of the manuscript.

==============================
Background

Nitrification can lead to large quantities of nitrate leaching into the soil during vegetable production, which may result in soil acidification in a greenhouse system. A better understanding is needed of the nitrification process and its microbial mechanisms in soil acidification.

Materials and Methods

A simulated acidification experiment with an artificially manipulated pH environment (T1: pH 7.0; T2: pH 6.5; T3: pH 6.0; T4: pH 5.5; T5: pH 4.5) was conducted in potted tomatoes grown in greenhouse conditions. The abundance and community structures of ammonia oxidizers under different pH environment were analyzed using q-PCR and high-throughput sequencing methods, respectively.

Results and discussions

Soil acidification was accompanied by a reduction of soil organic matter (SOM), total nitrogen (TN), NH3 concentration, and enzyme activities. The abundance of ammonia-oxidizing archaea (AOA) in the soil was higher than that of ammonia-oxidizing bacteria (AOB) in soils with a pH of 6.93 to 5.33. The opposite trend was observed when soil pH was 4.21. In acidified soils, the dominant strain of AOB was Nitrosospira, while the dominant strain of AOA was Nitrososphaera. The abundance and community structure of ammonia oxidizers were mainly affected by soil pH, NH4+ content, and microbial biomass. Soil nitrification activity (PNA) has a relationship with both AOA and AOB, in which the abundance of AOA was the crucial factor affecting PNA.

Conclusions

PNA was co-dominated by AOA and AOB in soils with simulated acidification. Changes of soil pH, NH4+, and microbial biomass caused by acidification were the main factors for the differences in the ammonia-oxidizing microbial community in greenhouse soils. Under acidic conditions (pH < 5), the pH significantly inhibited nitrification and had a strong negative effect on the production of tomatoes in greenhouse conditions.

Introduction

Tomato (Lycopersicon esculentum Miller) is a vegetable that can be eaten either fresh or processed, and its total annual output has always ranked first among vegetable crops in China. In 2018, the total greenhouse growing area of tomatoes was 642,000 hm2, accounting for 57.88% of the total tomato planting area (Ministry of Agriculture and Rural Affairs of China). A high multiple cropping index and the use of a large amount of fertilizer are common characteristics of tomato production systems in greenhouses; these practices cause a large accumulation of nutrients and the loss of soil nitrogen (Zhu et al., 2005; Fan et al., 2014; Bai et al., 2020). In addition, environmental factors such as high temperature, high humidity, semi-closed buildings, and no rainwater leaching into the greenhouse often lead to the formation of a unique soil acidification patterns during the production of greenhouse tomatoes (Han et al., 2014; Ju et al., 2007; Min et al., 2011; Zhou et al., 2010). Tomato is a nitrate-loving vegetable crop. Therefore, both ammonium nitrogen and organic nitrogen in the soil must be converted into nitrate nitrogen through the process of nitrification, which is conducive to the absorption and utilization by tomato plants. There are many recent studies on the effects of soil acidification on soil nitrification in farmland ecosystems (He et al., 2007; Yao et al., 2011; Wang et al., 2015a), but there are few reports on the characteristics and microbiological mechanisms of soil acidification on soil nitrification in intensive greenhouse vegetable production systems. It is important to clarify the response mechanism of soil nitrification to soil acidification in greenhouse conditions for the regulation of soil acidification and the improvement of soil nitrogen use efficiency in facility-grown tomato crops.

Nitrification performs a vital role in the biogeochemical cycle of nitrogen (Gruber & Galloway, 2008) and directly affects the soil nitrogen supply, and nitrogen absorption and utilization by plants. Ammonia oxidation, driven by ammonia-oxidizing bacteria (AOB) and archaea (AOA), is the first, rate-limiting step in the nitrification process (Kowalchuk & Stephen, 2001). The majority of prior research has shown that soil pH (Nicol et al., 2008; Bru et al., 2011; Gubry-Rangin et al., 2011; Hu et al., 2015) and substrate (ammonia/ammonium) (Martens-Habbena et al., 2009; Schleper, 2010) are the most important factors affecting the abundance and community compositions of both AOA and AOB. AOA was more competitive than AOB in a low-pH and low-ammonia environment owing to long-term niche differences (Wang et al., 2015a), whereas AOB typically grows in high ammonia concentrations (Verhamme, Prosser & Nicol, 2011). Increasing evidence has shown that AOB may dominate the autotrophic ammonia oxidation process in alkaline soils (Jia & Conrad, 2009; Xia et al., 2011), while AOA is the main ammonia oxidizer in acidic soils (Zhang et al., 2012; Yao et al., 2013; Wang et al., 2015a; Song et al., 2016). However, there were different perspectives that AOB featured in nitrification in acidic soils (Lin et al., 2021). In the acidic soil of tea gardens, the abundance of AOB had a positive correlation with pH changes, but AOA showed no correlation (Yao et al., 2011). This suggested that ammonia oxidizers have a more complex ecological provision and metabolic diversity in acidic soils.

Current research on the relationship of soil pH and nitrification has focused on the different types or amounts of fertilizer used. The application of chemical and organic fertilizers has been shown to change the soil physicochemical properties, soil nitrification potential, and the compositions of AOA and AOB (Chu et al., 2007; Schroder et al., 2011). The application of NPK fertilizers for 23 consecutive years resulted in soil acidification and enrichment of some AOA species. However, the abundance of AOA, AOB, and the potential nitrification activity increased significantly after the addition of organic fertilizers (Xue et al., 2016). A single application of nitrogen fertilizer led to a slight decrease in soil pH and the effects on ammonia oxidizers were different under other experimental conditions (Wessén et al., 2010; Tao et al., 2017). Some simulated acidification experiments have added an acid solution. In forest and grassland soils, the total and net nitrification rates were significantly stimulated by an elevated pH, whereas the soil nitrification rate was reduced due to the acidification (Cheng et al., 2013). In forest and farmland soils, the addition of different amounts of H2SO4 to generate pH gradients showed that the reduction in soil pH stimulated heterotrophic nitrification (Zhang et al., 2020). However, few experiments have explored the relationships between nitrification and acidification by simulating acidification in a greenhouse system.

A simulated acidification experiment was conducted in the greenhouse to determine the effect of soil acidification on nitrification and its microbial mechanism in the greenhouse system. We added different amounts of H2SO4 to neutral field soils to achieve different pH environment (pH = 7.0, pH = 6.5, pH = 6.0, pH = 5.5, pH = 4.5) in the potted tomato experiment. This study investigated the changes in soil properties, potential nitrification activity, the abundance and community structure of ammonia oxidizers under acidification conditions by using a shaken slurry, q-PCR, and high-throughput sequencing methods. The soil properties were correlated with ammonia oxidizers under simulated acidification conditions to evaluate the effects of soil acidification on an ammonia-oxidizing microbial community. The study of soil nitrification and its microbial mechanisms of acidified soils under greenhouse conditions provide a theoretical basis for the regulation of soil acidification and efficient nitrogen utilization in greenhouse tomato production.

Materials and Methods

Experimental design and soil sampling

The experimental site was located in the solar greenhouse of the Horticulture College research basement of Shenyang Agriculture University (41°82 N, 123°56 E) in Liaoning Province, China. The area features a typical temperate continental climate with an annual average temperature and precipitation of 8.4 °C and 658 mm, respectively. We chose neutral field soil and added different concentrations of H2SO4 for simulated acidification conditions in potted tomato plants. The chemical properties of the original soil were a pH of 7.21, soil organic matter (SOM) of 47.38 mg·kg−1, total nitrogen (TN) of 3.61 g·kg−1, available phosphorus (AP) of 317.68 mg·kg−1, and available potassium (AK) of 515.08 mg·kg−1.

We conducted five pH environmental treatments (T1: pH = 7.0; T2: pH = 6.5; T3: pH = 6.0; T4: pH = 5.5; T5: pH = 4.5). The corresponding acid solution concentration was 0.10, 0.13, 0.20, 0.53, and 1 ml·L−1. The potted experiment began in March 2021, and each treatment was repeated for 15 pots randomly placed in greenhouse experimental plots. The cultivation pot (30 cm diameter and 30 cm depth) was filled with 14 kg soil for one tomato plant, and a medium-fruit disease-resistant variety “Meisheng” tomato was planted. Before the start of the experiment, the soil was pre-mixed with an acid solution according to the pH gradient set in the experiment, and tomato planting was carried out after the soil of each treatment reached the desired pH gradient. The plants were watered with an acid solution once in the middle of two watering periods throughout the entire growth period of the tomato, and the amount of acid poured each time was kept constant. Before planting, 190 g of chicken manure was applied to each pot as a base fertilizer, and 15 g of NPK (15:15:15) compound fertilizer was applied to each pot after planting. Tomato plants were managed with single stem pruning, and plant height and stem diameter were measured 60 days after planting. When the tomato plant has set four flower clusters, the plant was pinched off at the growth point.

At the end of June 2021, tomato fruits were harvested after they were fully ripe. We added the weights of four ears as the yield per tomato plant. After harvesting the fruit, the roots, stems and leaves of the plants were dried to a constant weight in a drying oven at 80 °C. The dry weight of stems and leaves was considered to be the aboveground biomass and the dry weight of the roots was counted as the underground biomass; together this weight was calculated as the total biomass. After tomato plants were removed, we randomly collected three replicate rhizosphere soils (0–20 cm) from each treatment for analysis. Soil samples were collected in each pot from five positions using an auger and then were mixed thoroughly to form one sample. The soil samples were sieved with a 2 mm sieve after removing visible plants residues and were then divided into three portions. One portion was naturally dried for the determination of soil chemical properties, a second portion was stored at 4 °C until the soil ammonium nitrogen (NH4+-N), nitrate nitrogen (NO3−-N), microbial biomass, enzyme activity, and potential nitrification activity were analyzed, and a third portion was stored at −80 °C for later molecular analysis.

Soil chemical analysis

All soil chemical analyses refer to the methods used by Bao (2000). The soil pH was measured with a pH meter (pHS-25; INESA, Shanghai, China) in a 1:2.5 soil:water slurry using air-dried soil and CO2-free distilled water. SOM was determined by oxidative digestion with K2Cr2O7 and concentrated H2SO4 and then titrated with 0.5 M FeSO4 under the condition of an o-phenanthroline indicator. TN was digested with 5 ml concentrated H2SO4 and mixed catalyst at 350 °C and then analyzed by an automatic Kjeldahl nitrogen analyzer (BUCHI, Switzerland). Soil NH4+-N and NO3−-N were measured with a SAN++ continuous flow analyzer (Skalar, Breda, Netherlands) after extraction with 2 M KCl (soil: KCl = 1:10) for 1 h. The concentration of NH3 was calculated according to the formula: NH3 = NH4+*10(pH-pka), where pka = 9.245 at 25 °C (Norman & Barrett, 2014).

Soil microbial carbon (MBC) and microbial nitrogen (MBN) were determined by the chloroform fumigation-extraction method (Jenkinson & Powlson, 1976; Brookes et al., 1985; Vance, Brookes & Jenkinson, 1987). Briefly, three parts of 10 g fresh soil were weighed and placed in a glass dish, one part was fumigated with ethanol-free chloroform under 25 °C dark conditions for 24 h, while the other part was placed under the same conditions without chloroform. One part was used for the determination of the moisture content. The fumigated and unfumigated soil samples were then leached with 40 mL of 0.5 M K2SO4 at 220 rpm for 30 min. The solution was filtered through quantitative filter paper and analyzed by a TOC analyzer (Multi C/N 3100; Analytik Jena, Jena, Germany). Soil urease was measured using the phenol-sodium hypochlorite method, and soil protease was measured by the ninhydrin colorimetric method (Yan, 1988).

Potential nitrification activity (PNA)

Potential nitrification activity (PNA) was determined by the modified shaken slurry method (Xue et al., 2009). Briefly, 9 g of fresh soil was weighed into a 100 mL glass bottle, and 60 ml of phosphate buffer was added containing 1.5 mmol·L−1 (NH4)2SO4. An air-permeable sealing film was placed on top and the bottle was placed in a constant temperature shaking box at 25 °C for culturing. A total of 2 ml of soil homogenate was extracted at 2, 4, 6, 8, and 10 h respectively. The samples were then centrifuged to obtain the supernatant, and the nitrite nitrogen content in the solution was measured by colorimeter at 540 nm. The dynamic change trend of NO2−-N content was used as the linearity of ammonia oxidation and PNA was calculated as the production of NO2−-N per unit time.

Extraction of soil DNA and quantitative analysis of AOA and AOB

Soil DNA was extracted from 0.30 g of fresh soil using the FastDNA SPIN kit for soil (MP Biomedicals, Santa Ana, California, USA). DNA concentration and quality were determined with NanoDrop2000 (Thermo Fisher, Waltham, MA, USA). The abundance of the amoA gene of ammonia-oxidizing archaea (AOA) and ammonia-oxidizing bacteria (AOB) were determined by an ABI7300 real-time PCR system (Applied Biosystems, Waltham, MA, USA). The primers Arch-amoAF/Arch-amoAR and amoA-1 F/amoA-2R (Table 1) were used for quantifying the AOA and AOB amoA genes, respectively. The standard curve was made by referring to the method of He et al. (2007). Each DNA analysis was repeated three times, and the quantitative reaction system was 25 ul.

Table 1 Primers for quantitative analysis of amoA gene.

Target group	Primer	Sequence (5–3)	Real-time PCR conditions	Reference	
AOA	Arch-amoA F
Arch-amoA R	STAATGGTCTGGCTTAGACG
GCGGCCATCCATCTGTATGT	5 min at 95 °C, 35 cycles of 45 s at 95 °C, 30 s at 57 °C for AOB, and 60 s at 53 °C for AOA, 1 min at 72 °C	(Francis et al., 2005)	
AOB	amoA1 F
amoA2 R	GGGGTTTCTACTGGTGGT
CCCCTCKGSAAAGCCTTCTTC	(Rotthauwe, Witzel & Liesack, 1997)	

High-throughput sequencing and data processing

The same primers and conditions as quantitative analysis of AOA and AOB for amplification were used before high-throughput sequencing and the product was purified. The purified products were sequenced on an Illumina MiSeq PE300 platform (Illumina, San Diego, CA, USA). Before data analysis, the original data was spliced for quality control and optimized data, and operational taxonomic unit (OTU) clustering was performed according to a 97% sequence similarity. The complete data have been uploaded to the NCBI Sequence Read Archive (SRA) database under the accession numbers PRJNA837458 and PRJNA837646.

Statistical analysis

All statistical analysis was performed using the SPSS 18.0 (IBM, Armonk, NY, USA). One-way analysis of variance (ANOVA) followed by a Duncan’s least significant differences test was carried out to determine significant differences between the treatments, and p < 0.05 were considered statistically significant. Differences in microbial communities among all treatments were investigated using principal component analysis (PCA). In addition, permutation multivariate analysis of variance (PERMANOVA) based on the Bray-Curtis distance algorithm was used to represent significant differences in changes among all treatments. Pearson’s correlation coefficients were used to test the relationships among environmental factors (soil chemical properties and microbial activities), PNA, and ammonia oxidizers communities. Redundancy analysis (RDA) was conducted to represent the correlation of sample distribution and environmental factors. To evaluate the relative effects of environmental factors on PNA and ammonia oxidizing microbial communities, a random forest analysis method using a decision tree approach to assess the importance of variables was used. In the process of random forest calculation, we used the Shannon index to represent the ammonia-oxidizing microbial community diversity, and the PCA axis 1 to represent the ammonia-oxidizing microbial community composition. Random forest analysis can predict how much other variables explain the target variable and the importance of other variables to the target variable.

Results

Soil chemical analysis

Watering with different concentrations of acid solutions produced significantly different pH gradients and the soil pH values of each treatment were 6.93, 6.45, 5.93, 5.33, and 4.21, respectively (Table 2). Changes in soil pH were shown to significantly affect other soil chemical properties (p < 0.05) (Table S1). Soil SOM and TN content showed a decreasing trend with a decrease of soil pH. In T4 (pH = 5.33) and T5 (pH = 4.21) treatments, the soil SOM and TN content were the lowest and were significantly different from other treatments (p < 0.05). The content of NH4+-N ranged from 3.48 to 19.02 mg·kg−1, and there was an increasing trend as the soil pH decreased. In T4 (pH = 5.33) and T5 (pH = 4.21) treatments, the content of NH4+-N was significantly higher than in other treatments (p < 0.05). The NO3−-N content of the T5 (pH = 4.21) treatment was the lowest compared with other treatments (p < 0.05), and the differences between the other treatments were not significant. In addition, the highest NO3−-N content was detected in the T4 (pH = 5.33) treatment, at 25.95 mg·kg−1. The NH3 concentration showed a decreasing trend with the decrease of soil pH. Among the samples, the T1 (pH = 6.93) treatment had the highest NH3 concentration of 15.65 mg·m−3 (p < 0.05).

Table 2 Soil chemical properties of the different treatments.

Treatment	pH	SOM
(g·kg−1)	TN
(g·kg−1)	NH4+-N (mg·kg−1)	NO3−-N
(mg·kg−1)	NH3
(mg·m−3)	
T1	6.93 ± 0.02a	45.35 ± 0.57a	3.45 ± 0.04a	3.48 ± 0.10c	17.66 ± 2.77a	15.65 ± 1.74a	
T2	6.45 ± 0.03b	42.62 ± 1.03a	3.35 ± 0.04a	3.79 ± 0.23c	21.46 ± 4.14a	6.38 ± 0.79b	
T3	5.93 ± 0.01c	41.90 ± 1.91a	3.12 ± 0.05b	4.41 ± 0.42c	18.03 ± 1.26a	2.09 ± 0.19c	
T4	5.33 ± 0.02d	35.94 ± 0.63b	2.95 ± 0.05c	14.38 ± 1.10b	25.95 ± 3.93a	1.63 ± 0.08c	
T5	4.21 ± 0.01e	34.24 ± 0.79b	2.83 ± 0.06c	19.02 ± 1.14a	13.01 ± 4.70b	0.16 ± 0.02c	
Note:

Values are mean ± standard deviation (n = 3). Different letters indicate significant differences at 0.05 probability level. SOM (Soil organic matter), TN (Total nitrogen).

The effect of soil pH on microbial biomass and enzyme activities were shown in Fig. 1. The MBN and MBC of T5 (pH = 4.21) treatment were significantly lower than other treatments (p < 0.05). There was a very significant positive correlation between microbial biomass and NO3−-N content (p < 0.01). In addition, soil MBN also had a very significant positive correlation with soil pH and a very significant negative correlation with NH4+-N (p < 0.01) (Table S1). Soil urease and protease of T4 (pH = 5.33) and T5 (pH = 4.21) treatments were significantly lower than T1 (pH = 6.93), T2 (pH = 6.45), and T3 (pH = 5.93) treatments (p < 0.05). A correlation analysis of soil enzyme activities and soil chemical properties showed that soil urease and protease were positively correlated with soil pH, SOM, TN, and NH3 concentrations (p < 0.01), and negatively correlated with NH4+-N content (p < 0.01). Soil urease was also positively correlated with soil NO3−-N content (p < 0.05) (Table S1).

Figure 1 Soil microbial activity of the different treatments.

(A) Soil microbial nitrogen. (B) Soil microbial carbon. (C) Soil urease. (D) Soil protease. Lowercase letters at the top of the bar chart represent significant differences at 0.05 probability level.

Ammonia oxidation rate and PNA

The ammonia oxidation rate was expressed by the change of NO2−-N content with the incubation time, and the ammonia oxidation showed a linear relationship (Fig. 2). The slope magnitude of the linear relationship was ranked as T2 (pH = 6.45) (k = 2.249, R2 = 0.9861) > T1 (pH = 6.93) (k = 1.909, R2 = 0.9995) > T3 (pH = 5.93) (k = 1.476, R2 = 0.9995) > T4 (pH = 5.33) (k = 0.3387, R2 = 0.8897) > T5 (pH = 4.21) (k = 0.0752, R2 = 0.6382). The variation range of PNA for different treatments was 0.26–2.32 NO2−-N ug·g−1·h−1 (Fig. 2F). The highest and lowest PNA were detected in the T2 and T5 treatments, respectively, and the PNA of the T2 treatment was significantly higher than other treatments (p < 0.05). The correlation between PNA and the soil environmental factors is shown in Table S2. PNA has an extremely significant positive correlation with soil pH, SOM, TN, NH3 concentration, urease, and protease (p < 0.01), and an extremely significant negative correlation with NH4+-N content (p < 0.01). PNA was also shown to have a significant positive correlation with MBN (p < 0.05).

Figure 2 (A–F) Ammonia oxidation linear relationship and potential nitrification activity (PNA) of different treatments.

R2 demonstrates the linear relationship fit, the lowercase letters at the top of the bar chart represent significant differences at 0.05 probability level.

amoA gene abundance of AOA and AOB

amoA gene copy numbers were used to indicate the abundance of ammonia oxidizing bacteria (AOB) and archaea (AOA). The AOA and AOB amoA gene copies were in the range of 7.84 × 108 to 1.41 × 1010 and 4.53 × 108 to 1.15 × 109 copies g−1 d.w.s, respectively (Fig. 3). The ratios of AOA to AOB were in the range 0.62 to 18.46 among all treatments. The abundance of AOA was higher than that of AOB in each treatment, with the exception of the T5 (pH = 4.21) treatment. The abundance of AOA decreased with a decrease in soil pH; the copy number of AOA amoA genes of T4 (pH = 5.33) and T5 (pH = 4.21) treatments were significantly lower than other treatments. In the T1 (pH = 6.93) treatment, the copy number of the AOA amoA gene was 28.54 times that of the T5 (pH = 4.21) treatment. The highest and lowest AOB amoA gene copies appeared at T3 (pH = 5.93) and T5 (pH = 4.21), respectively, and the copy number of bacterial amoA genes in the T5 (pH = 4.21) treatment was significantly lower than the other treatments (p < 0.05).

Figure 3 Abundance of AOA and AOB amoA gene copy numbers under different treatments.

amoA gene copy numbers were log-transformed, and the different letters indicate significant differences at 0.05 probability level (capital letters indicate the significance of AOA, lowercase letters indicate the significance of AOB). The ratios of AOA to AOB were shown in the boxes above the chart.

Community diversity and composition of AOA and AOB

The sequencing results of the AOA and AOB communities were shown in Table 3. The AOA and AOB of each sample had 8,222 and 8,244 reads, respectively, once all of the sequencing results were standardized. The coverage value of AOA and AOB were both above 0.99, indicating that the sequencing depths of all samples were sufficient and the sequencing data was reasonable. ACE and Chao1 indices were used to represent the richness of species, and the Shannon index was used to characterize species diversity. For the AOA community, the number of OTUs in the T5 (pH = 4.21) treatment was significantly lower than other treatments (p < 0.05). Likewise, its richness was also significantly lower than other treatments. The diversity of T4 (pH = 5.33) treatment was significantly higher than other treatments (p < 0.05). As the soil pH decreased, the number of OTUs showed a decreasing trend for AOB. The richness of the T4 (pH = 5.33) and T5 (pH = 4.21) treatments were significantly lower than other treatments (p < 0.05). The diversity of the T1 (pH = 6.93) treatment was significantly higher than that of other treatments (p < 0.05), while the diversity of the T5 (pH = 4.21) treatment was significantly lower than that of the other treatments (p < 0.05).

Table 3 Illumina Miseq sequencing results and alpha diversity analysis.

Treatment	Sequencing results	Richness	Diversity	Coverage	
Read	OTUs	ACE	Chao1	Shannon	
AOA community	T1	8222	61a	69.20 ± 0.63a	78.96 ± 6.34a	1.50 ± 0.12b	0.9987	
T2	8222	62a	72.19 ± 3.46a	70.49 ± 2.04a	1.31 ± 0.06b	0.9985	
T3	8222	59a	67.28 ± 3.27a	67.00 ± 4.04a	1.69 ± 0.05b	0.9988	
T4	8222	68a	73.71 ± 1.48a	77.99 ± 2.39a	2.32 ± 0.02a	0.9989	
T5	8222	33b	35.98 ± 11.11b	34.50 ± 10.77b	1.70 ± 0.36b	0.9996	
AOB community	T1	8244	59a	63.98 ± 1.25a	61.04 ± 0.87a	2.63 ± 0.01a	0.9993	
T2	8244	62a	66.67 ± 1.47a	64.72 ± 1.66a	2.38 ± 0.07b	0.9992	
T3	8244	49b	61.69 ± 5.48a	58.53 ± 3.79a	2.04 ± 0.08c	0.9990	
T4	8244	39c	41.93 ± 2.63b	42.17 ± 3.35b	2.20 ± 0.03bc	0.9994	
T5	8244	29d	39.61 ± 11.05b	37.15 ± 10.23b	1.12 ± 0.10d	0.9993	
Note:

Values are mean ± standard deviation (n = 3). Different letters indicate significant differences at 0.05 probability level.

All AOA OTUs were assigned into four phyla and six genera, Thaumarchaeota was the dominant group with a proportion of 10.69–83.60% in all treatments at the phylum level, and the proportion of Thaumarchaeota in T5 (pH = 4.21) treatment was the smallest (Fig. 4A). Among six species at the genus level only one species was named, namely Nitrososphaera (Fig. 4B). The relative abundance of Nitrososphaera was the highest in the T4 (pH = 5.33) treatment, accounting for 46.87%, followed by the T3 (pH = 5.93) treatment, accounting for 15.52%, which was significantly higher than the other treatments (p < 0.05) (Fig. 4C). For the AOB community, the OTUs were assigned into three phyla and six genera. At the phylum level, Proteobacteria had the largest relative abundance, and the proportion of all treatments accounted for 76.11–95.10% (Fig. 4D). Among six species at the genus level, only Nitrosospira was named (Fig. 4E). Among these species, this specie had the largest overall proportion in the T5 (pH = 4.21) treatment, accounting for 88.05%, which was significantly higher than other treatments (p < 0.05) (Fig. 4F).

Figure 4 Relative abundance of ammonia-oxidizing archaea (AOA) and bacteria (AOB) at phylum and genus level.

(A) Relative abundance of AOA at the phylum level. (B) Relative abundance of AOA at the genus level. (C) Relative abundance of dominant AOA taxa (nitrososphare). (D) Relative abundance of AOB at the phylum level. (E) Relative abundance of AOB at the genus level. (F) Relative abundance of dominant AOB taxa (nitrosospira).

PCA was used to analyze the differences in AOA and AOB community composition among different treatments based on the Bray-Curtis distance matrix. PCA showed that the two axes explained 67.58% of the total variation in the AOA community (PERMANOVA: R = 0.5378, p = 0.001). The AOA community composition of the T4 (pH = 5.33) and T5 (pH = 4.21) treatments were clearly separated from other treatments on the PCA axis 1 (Fig. 5A). Similarly, PCA showed that the two axes explained 100% of the total variation of the AOB community composition (PERMANOVA: R = 0.8874, p = 0.001). The AOB community composition of the T3 (pH = 5.93) and T4 (pH = 5.33) treatments were clearly separated from other treatments on the PCA axis 1 (Fig. 5B).

Figure 5 Principal component analysis (PCA) of ammonia-oxidizing archaea (AOA) and bacteria (AOB) based on the Bray-Curtis distance matrix at the genus level.

(A) PCA of AOA. (B) PCA of AOB. The annotations on the top of the figure represent the significance of differences.

Correlation between the abundance, diversity and composition of ammonia oxidizers with environmental factors

The correlations between the abundance and diversity of ammonia oxidizers and environmental factors are shown in Table S2. The abundances of AOA and AOB were both significantly positively correlated with soil pH, SOM, TN, MBN, urease, and protease (p < 0.05), and significantly negatively correlated with NH4+-N content (p < 0.01). Additionally, the AOA abundance was also significantly positively correlated with the NH3 concentration (p < 0.01). The diversity of AOB was significantly correlated with all of the measured environmental factors (p < 0.05), however, the diversity of AOA was not. The relationships between the community compositions of AOA or AOB and environmental factors were tested using RDA analysis (Fig. 6). The results showed that the x-axis and y-axis can explain 64.82% and 67.79% of the total variation of AOA and AOB community composition, respectively. Soil pH, SOM, TN, NH4+-N, NH3, urease, and protease had longer projection lengths on the first ordinal axis of AOA and AOB, respectively, indicating that they had a significant impact on community composition (Table S4).

Figure 6 Distance-based redundancy analysis (db-RDA) of ammonia-oxidizing archaea (AOA) and bacteria (AOB).

(A) db-RDA of AOA. (B) db-RDA of AOB. In the graph, an asterisk (*) indicates the p-value of correlation; *0.01 < p ≤ 0.05, **0.001 < p ≤ 0.01, ***p ≤ 0.001.

Relative importance of environmental factors on PNA and ammonia oxidizing microbial communities

Random forest analysis was used to calculate the relative importance of each environmental factor to the differences in the ammonia-oxidizing microbial community in order to assess the relative impact of environmental factors on the ammonia-oxidizing microbial community. The model showed that NH4+-N content explained the most to the abundance and composition of AOA (Figs. 7A and 7E). The soil pH has the greatest impact on AOB abundance and diversity of all the environmental factors (Figs. 7B and 7D). MBN and MBC exerted a strong effect on the diversity of AOA (Fig. 7C) and the composition of AOB (Fig. 7F), respectively. In addition, the results showed that the soil parameter with the greatest effect on PNA was the soil pH (Fig. 7G); the abundance of AOA had the greatest effect on the PNA (Fig. 7H). The data from the correlation analysis showed that PNA was significantly associated with the abundance, diversity, and community composition of ammonia oxidizers (Table S3).

Figure 7 Relative variable importance of environmental factors for the ammonia oxidizing microbial community.

(A) Abundance of AOA. (B) Abundance of AOB. (C) Shannon index of AOA. (D) Shannon index of AOB. (E) PCA axis1 of AOA. (F) PCA axis1of AOB. (G and H) Potential nitrification activity (PNA).

Effects of simulated acidification on tomato growth and yield

The effect of long-term acidification on the growth of potted tomato is shown in Table 4. The plant height of the T2 (pH = 6.45) and T4 (pH = 5.33) treatments were significantly higher than that of the T1 (pH = 6.93) treatment. However, the differences in stem diameter among the treatments were not significant. The total biomass of the T4 (pH = 5.33) treatment was the highest, and the total biomass of T5 (pH = 4.21) was significantly lower than the other treatments (p < 0.05). The aboveground biomass of the T4 treatment was 1.3 times that of the T5 treatment, and the underground biomass of the T4 treatment was 2.18 times that of the T5 treatment. The yield analysis of all acidification treatments showed that the pH of T5 (pH = 4.21) resulted in a significantly lower yield than other treatments (p < 0.05) (Table 4). The correlation of the total biomass and yield with environmental factors was shown in Table S2. The total biomass of the tomato plant was significantly positively correlated with soil nitrate nitrogen content and microbial biomass (p < 0.01). In addition to being affected by pH, the yield was also significantly positively correlated with soil NO3−-N content, microbial biomass, and enzyme activity (p < 0.01). Ammonium nitrogen content was negatively correlated with tomato yield (p < 0.05).

Table 4 Tomato plant growth indexes of different treatments.

Treatment	Plant height
(cm)	Stem diameter (cm)	Aboveground biomass (g/plant)	Underground biomass (g/plant)	Total biomass (g/plant)	Yield
(kg/plant)	
T1	158.33 ± 0.88b	1.08 ± 0.08a	189.52 ± 4.15d	5.42 ± 0.27b	194.60 ± 4.06c	1.98 ± 0.02a	
T2	165.00 ± 1.00a	1.11 ± 0.01a	201.92 ± 4.49c	8.60 ± 0.28a	209.93 ± 4.11b	1.97 ± 0.04a	
T3	162.67 ± 0.33ab	1.20 ± 0.02a	213.49 ± 3.64b	7.46 ± 0.61a	221.17 ± 5.11b	2.12 ± 0.03a	
T4	166.33 ± 4.91a	1.25 ± 0.09a	242.10 ± 2.95a	8.06 ± 0.80a	250.10 ± 2.10a	2.10 ± 0.08a	
T5	164.33 ± 0.88ab	1.17 ± 0.04a	173.34 ± 2.01e	3.69 ± 0.09c	177.05 ± 2.10d	1.63 ± 0.03b	
Note:

Values are mean ± standard deviation (n = 3). Different letters indicate significant differences at 0.05 probability level.

Discussion

Effects of acidification on AOA and AOB abundance

The abundances of AOA and AOB decreased with lower soil pH levels and were significantly positively correlated with soil pH (Fig. 3, Table S2), which was similar to the results of previous studies (Shen et al., 2008; Chen et al., 2013). Random forest analysis also indicated that the most important factor affecting AOB abundance was soil pH (Fig. 7B). The abundance of AOA was higher than that of AOB (Fig. 3) when the soil pH was 6.93-5.33, which was consistent with previous reports that AOA was dominant in acidic soils (Leininger et al., 2006; Di et al., 2009; Ai et al., 2013). However, when pH = 4.21, the abundance of AOB was higher than AOA (Fig. 3). This may be due to the long-term acidification, which caused acid-resistant strains appear in AOB (Li et al., 2018; Picone et al., 2021), and the presence of acid-tolerant strains resulted in a higher abundance of AOB than AOA.

In addition to the pH, ammonia concentration was considered to be a major factor driving the ammonia oxidizing microbial community and abundance (Prosser & Nicol, 2012; Yao et al., 2013). The availability of ammonia was considered to be an important factor in inducing the growth and niche of AOA and AOB (Erguder et al., 2009; Martens-Habbena et al., 2009). The different responses of AOB and AOA to pH may be due to their respective affinities for ammonia substrate (Levičnik-Höfferle et al., 2012). We found that a decrease of soil pH resulted in an increase in NH4+ (Table 2), and a decrease in the AOA abundance (Fig. 3). The AOA abundance was significantly negatively correlated with NH4+ content (Table S2). The relative importance analysis showed that NH4+-N content was the most important factor affecting the abundance of AOA (Fig. 7A). This may be due to the change in the balance between NH3 and NH4+ as a result of a decrease of the soil pH. This, in turn, led to an increase in NH4+, which inhibited the growth of AOA and reduced the activity and abundance sensitivity of AOA (Boer & Kowalchuk, 2001; Verhamme, Prosser & Nicol, 2011). The abundance of AOB was lower with a decreasing pH, which may be due to a reduced NH3 concentration as a result of lower soil pH (Table 2). AOB was shown to prefer a high ammonia environment (Norman & Barrett, 2014). When the soil pH was 4.21, the abundance of AOB decreased less than that of AOA, possibly because there were AOB species with a greater tolerance to soil NH4+ compared with AOA (Prosser & Nicol, 2012). Some studies have also shown that the increase of NH4+ will lead to the increase of AOB quantity and nitrification activity (Okano et al., 2004; Dang et al., 2018).

Effects of acidification on the community structure of AOA and AOB

Soil pH has effects on the diversity of ammonia-oxidizers (Nicol et al., 2008). We showed that the Shannon index of AOA increased with decreasing of soil pH (Table 3), which was in contrast to the previous finding that AOA diversity increased with a higher soil pH (Gubry-Rangin et al., 2011; Pester et al., 2012). This may be due to the increase of NH4+ with a lower soil pH (Table 2); AOA can grow in NH4+ rich conditions and exhibit a high degree of diversity (Francis et al., 2005). The Shannon index of AOB in this study was significantly positively correlated with soil pH (Table S2), and random forest analysis suggested that the AOB diversity was primarily ascribed to the soil pH (Fig. 7D). These results are consistent with previous findings that the pH was the main factor affecting AOB diversity (Guo et al., 2017).

Soil pH is a key explanatory variable for the variation in community structure of AOA and AOB (Yao et al., 2011; Zhou et al., 2014), and our results confirmed this. The community compositions of AOA and AOB had different responses to pH, which can be seen from the PCA results (Fig. 5). The AOA community composition of the T4 and T5 treatments were separated from other treatments on the PCA1 axis (Fig. 5A), and the AOB community composition of the T3 and T4 treatments were separated from other treatments on the PCA1 axis (Fig. 5B). The random forest analysis showed that the NH4+-N content and MBN were the most important factors affecting the composition of AOA and AOB communities, respectively. Soil acidification led to changes in the chemical properties and microbial biomass of the soil, which affected the composition of ammonia-oxidizing microbial communities.

Under simulated acidification conditions, Nitrososphaera was the main AOA among the different treatments (Fig. 4E). This was in contrast to previous studies in which the dominant AOA cluster in acidic soils was Nitrosotalea (Lehtovirta-Morley et al., 2011; Lu et al., 2012; Wang et al., 2015b). However, active AOAs were also reported to belong to the Nitrosotalea and Nitrososphaera clusters in five strongly acidic soils (pH < 4.5) (Zhang et al., 2012). A recent study provided strong evidence for the adaptive growth of Nitrososphaera-like AOA in acidic soil (pH = 4.92) (Wang et al., 2014). Tourna et al. (2011) showed that most cultivable Nitrososphaera are neutrophilic, and the nitrification activity of these AOA was significantly reduced or absent at pH values below 5.5. Similarly, it was also observed that when the pH was 5.93, Nitrososphaera dominated the populations, and when the pH was 4.21, it made up the smallest population. Nitrosospira was the main AOB in different acidification treatments (Fig. 4F), which was consistent with most studies (Avrahami & Conrad, 2003; Chen et al., 2011). In acidic soils, Nitrosospira Cluster 2, Nitrosospira Cluster 3, and Cluster 9 are often found to be the main active AOB (Kowalchuk & Stephen, 2001; Wang et al., 2015a). In our study, Nitrosospira accounted for the highest proportion in the T5 (pH = 4.21) treatment, which may be related to the previous report that Nitrosospira cluster 3a.2 had higher nitrogen (perhaps NH4+-N) demand (Avrahami, Conrad & Braker, 2003).

The contribution of AOA and AOB to nitrification under the acidification soils

Controversies exist regarding whether AOA or AOB dominates nitrification in acidic soil (Boer & Kowalchuk, 2001; Li et al., 2018). Many studies claimed that AOA plays a more important role than AOB in autotrophic ammonia oxidation in strongly acidic soils (Zhang et al., 2012). Studies have also shown that the AOB community dominates the nitrification process (Di et al., 2009), that the abundance of AOB has a significant relationship with the nitrification rate (Xia et al., 2011), and that AOB dominates ammonia oxidation in specific acidic soils ( Huang et al., 2018; Lin et al., 2021). In our study, PNA was significantly positively correlated with AOA and AOB abundance, AOB diversity, and AOA community composition, but was significantly negatively correlated with AOA diversity (Table S2). These results indicate that nitrification was driven by both AOA and AOB under long-term acidification conditions in greenhouse systems (Fig. 8). The AOA abundance was the most important factor affecting PNA according to random forest analysis (Fig. 7H).

Figure 8 Summary of this study.

The arrows represent the correlation between the two indicators.

Our research showed that soil pH and PNA had a very significant positive correlation under long-term acidification conditions (r = 0.888, p = 0.000) (Table S2), which was consistent with the results of previous studies on the effect of acidification on nitrification (He et al., 2007). It was observed that when the pH decreased from 6.93 to 6.45, the PNA increased 1.18 times (Fig. 2F). The same report showed that in acidic soils, the nitrification rate increased 4.6 times when the soil pH decreased from 6.2 to 5.7 (Zhu et al., 2011). A soil pH of 6.93 resulted in a lower PNA, which may be due to high NH3 concentrations that limited the activity of the ammonia oxidant. The decrease in pH led to a decrease in the PNA, which validated previous findings on acidification in acidic forests and grasslands, reducing the soil nitrification rate (Cheng et al., 2013). Under long-term acidification conditions, the decrease of PNA will affect the supply of soil nitrogen and the absorption and utilization of nitrogen by crops, thereby affecting the growth of crops. In the process of greenhouse tomato production, the application of acidic fertilizers should be avoided to ensure the normal growth of crops, and the acid-base balance of the soil can be maintained by adding organic fertilizers or some alkaline materials.

Conclusions

Acidification led to changes in soil chemical properties, and affected microbial activity and the abundance and community structures of AOA and AOB involved in ammonia oxidation. The PNA of acidified soil was co-dominated by AOA and AOB, and the abundance of AOA has the greatest effect on the PNA. The effect of acidification on soil nitrification and ammonia oxidizers was dependent on the pH, and substrate and microbial biomass changes caused by acidification. Under acid-neutral conditions (pH = 6.93–5.33), plant growth would not be significantly affected. A pH less than five resulted in weaker nitrification activity, which adversely affected the yield of tomatoes in greenhouse.

Supplemental Information

Supplemental Information 1 Person Correlations between the environmental factors.

In the table, an asterisk (*) indicates the p-value of correlation, and the number indicates the R-values of correlation; * 0.01 < p ≤ 0.05, ** 0.001 < p ≤ 0.01.SOM (Soil organic matter), TN (Total nitrogen), MBC (Soil microbial carbon), MBN (Soil microbial nitrogen).

Click here for additional data file.

Supplemental Information 2 Correlations of environmental factors with PNA, ammonia-oxidizers abundance, diversity, and tomato plant growth.

In the table, an asterisk (*) indicates the p-value of correlation, and the number indicates the R-values of correlation; * 0.01 < p ≤ 0.05, ** 0.001 < p ≤ 0.01.SOM (Soil organic matter), TN (Total nitrogen), MBC (Soil microbial carbon), MBN (Soil microbial nitrogen), PNA (Potential nitrification activity).

Click here for additional data file.

Supplemental Information 3 Pearson’s correlation analyses of PNA and AOA, AOB abundance, and community structure.

Diversity was represented by the Shannon index and community composition was replaced by the PCA1 axis. In the table, an asterisk (*) indicates the p-value of correlation, and the number indicates the R-values of correlation; * 0.01 < p ≤ 0.05, ** 0.001 < p ≤ 0.01. PNA (Potential nitrification activity).

Click here for additional data file.

Supplemental Information 4 RDA Envfit environment factors table of AOA and AOB.

Click here for additional data file.

Supplemental Information 5 Raw data for Table 2.

Click here for additional data file.

Supplemental Information 6 Raw data for Figure 1.

Click here for additional data file.

Supplemental Information 7 Raw data for Figure 2.

Click here for additional data file.

Supplemental Information 8 Raw data for Table 4.

Click here for additional data file.

Supplemental Information 9 Sequencing data of ammonia-oxidizing archaea with T1 treatment.

Click here for additional data file.

Supplemental Information 10 Sequencing data of ammonia-oxidizing archaea with T2 treatment.

Click here for additional data file.

Supplemental Information 11 Sequencing data of ammonia-oxidizing archaea with T3 treatment.

Click here for additional data file.

Supplemental Information 12 Sequencing data of ammonia-oxidizing archaea with T4 treatment.

Click here for additional data file.

Supplemental Information 13 Sequencing data of ammonia-oxidizing archaea with T5 treatment.

Click here for additional data file.

Supplemental Information 14 Sequencing data of ammonia oxidizing bacteria for T1 treatment.

Click here for additional data file.

Supplemental Information 15 Sequencing data of ammonia oxidizing bacteria for T2 treatment.

Click here for additional data file.

Supplemental Information 16 Sequencing data of ammonia oxidizing bacteria for T3 treatment.

Click here for additional data file.

Supplemental Information 17 Sequencing data of ammonia oxidizing bacteria for T4 treatment.

Click here for additional data file.

Supplemental Information 18 Sequencing data of ammonia oxidizing bacteria for T5 treatment.

Click here for additional data file.

Thanks to Majorbio Bio-Pharm Technology Co. Ltd. (Shanghai, China) for the sequencing service and Mr. Xiao Li for his help in sampling.

Additional Information and Declarations

Competing Interests

Author Contributions

Data Availability

The authors declare that they have no competing interests.

Xiaolan Zhang conceived and designed the experiments, performed the experiments, analyzed the data, prepared figures and/or tables, authored or reviewed drafts of the article, and approved the final draft.

Xuan Shan analyzed the data, prepared figures and/or tables, and approved the final draft.

Hongdan Fu conceived and designed the experiments, authored or reviewed drafts of the article, and approved the final draft.

Zhouping Sun conceived and designed the experiments, authored or reviewed drafts of the article, and approved the final draft.

The following information was supplied regarding data availability:

The raw data are available in the Supplemental Files.

The sequences are available at NCBI Sequence Read Archive (SRA): PRJNA837458 and PRJNA837646.

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
