# Peer review of "Effects of artificially-simulated acidification on potential soil nitrification activity and ammonia oxidizing microbial communities in greenhouse conditions"

_PeerJ, doi:10.7717/peerj.14088_

## Round 0.1 · original submission · Major Revisions

The manuscript needs major revision.

Reviewer 1 ·

Basic reporting

This manuscript have reviewed the literature on the effects of acidity on the nitrification in the soil. The Enlgish is clear and the field backgroud were provided. The structure is clear and rational. The figures and tables were made clear. All the contents in the MS were logistic.
The language and grammar need to revise, eg. P73, in different conditions, the effect should be effects, and the was –were

The sentence of Lines 100-101 is not right. Eg. …the soil were, PH 7.21…
Line 155-156 the sentence is wrong, please to check it.
L218 Nitrite nitrogen should be consistent with the other expression, eg. NO3–—N, NH4+-N。

Experimental design

This article is focused on the effects of acidication in the soil on the nitrification activity and ammonia oxiding microbes. The aims is distinct and it lies in the field of soil microoganism on the the effects of nitrification. The results were significant for the management of N fertilizer. The experiment was designed scientifically. The data is accurate.
The methods on the treatment of soil pH values were not clear. The reader cant know the step. Please clarify the detailed procedures on acidity you have obtained.

Validity of the findings

The findings were significant for the nitrogen deposition management. Thus, the results obtained in article were complete, so the there are strong novelty in this article.

Additional comments

NO.

Reviewer 2 ·

Basic reporting

Line 41: large accumulation (of what?)
Line 45-47: English and logic in this part need to be improved.
Line 48: Delete ‘greatly’
Line 54: Delete ‘Furthermore, ‘. Please pay attention to the use of conjunctive adverb in this article. They should be used only if there were logical connections between clauses.
Line 61: In acidic soil of tea gardens,
Line 100: Please explain the abbreviations when they appear in the manuscript for the first time, for example, SOM, TN, AP, AK, OUT, etc. Then you do not have to repeat to explain the abbreviation, such as PNA in lines 149 and 183.
Lines 115 and 117: Ears? Do you mean trusses? Did you weigh the whole shoots as yield instead of tomato fruit? Why use tomato fruit as yield?
Line 136-137: The font needs to be unified.
Line 164: DNA analysis

Experimental design

Line 105: What experimental design was used in this study?
Line 139-147: You need to cite references for those chemical analysis methods, including but not limited to chloroform fumigation-extraction method, phenol-sodium hypochlorite method, and ninhydrin colorimetric method. If any changes were made during the operation of those methods, it need to be clarified. Any lab equipment involved in these analyses need to be mentioned.
Line 172-173: I did not find data in SRA database via the accession numbers you provided.
Table 4: Why n=3? It was mentioned there were 15 plants included in each treatment in line
105.

Validity of the findings

Table 2: Please add legend explaining the abbreviations used in this table.
Table S1: There was no legend for Table S1. Please explain the meanings of various numbers of asterisk. Similar problems can be found in Table S2 and S3.
In Figure 5, PCA was based at genus level. In Figure 7 E&F, PCA axis 1 of AOA & AOB were explained by soil chemical properties and microbial activity. Please list descriptors/variables used in PCA in Methods. Besides, in lines 269 to 277, please explain what axis 1 represents in PCA of AOA, and AOB, respectively.
Figure 7: Please capitalize letters a-h since you use capitalized letters in legend and descriptions in manuscript.

Additional comments

The English language can be improved through finding a colleague who is proficient in English to review your manuscript or contact a professional editing service.
I would appreciated if each comment will be replied by authors through a reponse letter.

Reviewer 3 ·

Basic reporting

The authors investigated the correlation between pH environments and the potential nitrification activity of soil (by ammonia oxidizing microbes) during the growth of tomatoes in greenhouse. The authors have applied a variety of analytical methods to determine the concentration of nitrogen-containing species, level of enzymatic activities, the abundance and the sequence of microbes (AOA and AOB) as a result of vegetable growth in different pH environments.

Ultimately this work could be suitable for publication in PeerJ, following major revisions relating to the following -
1) Justification on the novelty of this work in the introduction -
In lines 65-81, the authors have outlined prior work on studying the relationship between soil pH and the nitrification process and mentioned that most studies were done in the field (e.g., in forests, grasslands, and farmlands). The authors therefore chose to study this relationship in greenhouse. It is unclear (at least in the introduction) how the environment in greenhouse is unique to the extent that it will lead to different findings compared to prior work.
The authors could discuss key differences between the greenhouse environment compared to the environment on the field; why the greenhouse environment could lead to different findings compared to previous studies on the field; what is your hypothesis on the proposed greenhouse study?

2) Justification of the significance of this work both in the introduction and in the discussion section -
In the introduction, I recommend the authors to comment on the percentage of vegetables grown in greenhouse compared to on the field (e.g., distribution and trends; using references) to justify the importance and relevance of conducting the proposed greenhouse study.
In the discussion section, I recommend the authors to further apply their findings to the current (and most common) growing conditions of vegetables (e.g., tomatoes) in greenhouse, what are the implications of their findings? E.g., What are implications of “acid rain” on vegetable growth, and can you make any recommendation based on this work? Should farmers “neutralize” the water before watering their plants?

Experimental design

3) Justification on the choice of species -
Why tomato? If growing a different type of vegetable, would the authors expect similar or different findings?

4) pH control (and tracking) during the experiment
Although the Initial pH of the soil and the pH of the watering solution was controlled by adding different amounts of sulfuric acid, I wonder if the authors have tracked how the pH of the soil may have changed over time (over multiple watering cycles and over the growth period of the tomatoes)? This is important for understanding the implication of this study, as pH values may drift or fluctuate over time. The authors could add this information to both the Methods section and the Results section.

Validity of the findings

5) Unclear speciation of nitrogen-containing species
• The authors have discussed the “Nitrification” process as turning ammonia into nitrate and nitrite. How about turning ammonia into dinitrogen (N2) and other nitrogen-containing gases (e.g., NO, NO2, and N2O)?
• Lines 49-50 “the ammonia oxidation… is the first and the rate-limiting step” – could the authors comment on the oxidation product here? Is it nitrite, or nitrate or other nitrogen-containing species?
• Lines 134-135: “soil inorganic nitrogen” – do the authors mean ammonium, or nitrate or nitrite, or all inorganic nitrogen-containing species combined?
• In Line 126, the authors mentioned ammonium and nitrate, but how about nitrite NO2-?
• In Figure 2, the authors reported nitrite concentrations as a function of time. It is unclear why only nitrite is reported here rather than nitrate (or nitrite and nitrate combined).

Additional comments

Minor comments:

6) Throughout the manuscript, the authors have used the term “pH gradient”, which means there is a change in pH within a medium. The authors perhaps want to use the term “pH environment” or “pH value” instead.

7) Method citations: please add citations if some of the analytical methods have been used and demonstrated in previous nitrification studies

8) I understand that the authors may not be fluent English speakers. I therefore recommend this manuscript to go through language editing services.

The authors have chosen an interesting topic of research, and they have done an extensive amount of experiment and data analysis. I recommend the authors to i) clarify the novelty and significance of the work, ii) justify the methods used in this study, iii) explain the speciation of nitrogen-containing species, and iv) improve the grammar prior to Acceptance.

---

## Round 0.2 · accepted · Accept

The manuscript is improved and accepted for publication.

Reviewer 3 ·

Basic reporting

The authors have thoroughly addressed all comments.

Experimental design

The authors have thoroughly addressed all comments.

Validity of the findings

The authors have thoroughly addressed all comments.

Additional comments

The authors have thoroughly addressed all comments.